# The Microwave Facile Synthesis of NiOₓ@graphene Nanocomposites for Application in Supercapacitors: Insights into the Formation and Storage Mechanisms

Yue Liang [1], Zhen Wei [2], Ruigang Wang [2,*] and Xinyu Zhang [1,*]

1 Department of Chemical Engineering, Auburn University, Auburn, AL 36849, USA; yzl0242@auburn.edu
2 Department of Metallurgical and Materials Engineering, The University of Alabama, Tuscaloosa, AL 35487, USA; zwei17@crimson.ua.edu
* Correspondence: rwang@eng.ua.edu (R.W.); xzz0004@auburn.edu (X.Z.)

**Abstract:** Recently, the strategy of combining carbon-based materials with metal oxides to enhance the electrochemical performance of electrodes has been a topic of great interest, but research focusing on the growth and charge storage mechanisms of such hybrid electrodes has rarely been conducted. In this work, a simple, reproducible, low-cost, and fast microwave heating method was used to synthesize NiOₓ@graphene nanocomposites. NiOₓ@graphene nanocomposites were used as a model system for exploring the growth and charge storage mechanisms of the hybrid electrode materials due to their simple preparation process, good stability, low cost, and high specific capacitance. The effects of reaction conditions (the type of metal precursor and feeding ratio between the nickel precursor and graphene) on the formation mechanism of the electrodes were examined, and it was demonstrated that the microstructure and morphology of the electrode materials were metal precursor-dependent, which was directly related to the electrochemical performance of the electrodes. Our work provides a new affordable approach to the synthesis of, and experimental support for designing, hybrid electrode architectures with a high electrochemical performance for next-generation energy storage devices.

**Keywords:** microwave heating method; nanocomposites; supercapacitor

## 1. Introduction

With the rapid evolution of the world that has led to the gradual exhaustion of fossil fuels, global warming, and pollution issues, there is an urgent need to develop a low-cost, environmentally friendly, and safe electrochemical energy storage system (EESS) to support the progress of low-carbon or zero-carbon sustainable economies in the future. A supercapacitor (SC) is a kind of energy storage device that combines the electrochemical properties of a high energy density battery and power density capacitor [1,2]. Compared to conventional energy storage systems (batteries, capacitors, and fuel cells), SCs are promising candidates for energy storage, due to their fast delivery rate, low cost, lack of memory effect, long life cycle, and the fact that they are lightweight and environmentally safe, which makes them irreplaceable in many portable systems and hybrid electric vehicles [3,4].

As the core of the SC, the electrode materials occupy a key position and directly determine the electron transport and electrochemical storage characteristics of the SC [5,6]. Normally, there are three types of electrode materials, including carbon-based materials, metal oxides, and conductive polymers. The carbon-based materials have the advantages of wide availability, a large specific surface area, and a wide operating temperature range. Metal oxides are regarded as a key electrode material due to their properties of simple preparation, high theoretical capacitance, non-toxicity, and good thermal and chemical stability. Unfortunately, their propensity for agglomeration at high mass loading, poor rate capability, weak electrochemical stability, and low conductivity properties hinder their further development. Conductive polymers with high conductivity and good mechanical

properties can effectively compensate for the shortcomings of metal oxides, but their specific capacitance is too low, with a short charge–discharge life cycle. To address these problems, the use of a hybrid electrode material combining different types of materials to advance the electrochemical performance of electrodes has recently been widely proposed. For instance, the multifunctional composite materials of NiO/CNT with an interlinked porous structure were formed using a simple wet chemical method followed by thermal annealing. This composite demonstrated a high supercapacitive performance (878.19 F/g at 2 mV/s) and excellent activity in the oxygen evolution reaction. The device, known as NiO-CNT//Activated Carbon, exhibited a specific energy of 85.7 Wh/kg at a power density of 11.2 kW/kg [7]. The CoO@polypyrrole electrode was prepared by anchoring the polypyrrole to the CoO nanowire, leading to a high specific capacitance (2223 F/g) and power density (5500 W/kg at 11.8 Wh/kg). The pseudocapacitive performance boost of the electrode was derived from the synergistic effect between the mesoporous CoO nanoparticles and the high-conductive PPy [8]. Monolayer graphene/NiO nanosheets were fabricated by Jiang et al. in 2011. The hybrid structure of the composite prevented the aggregation of the NiO nanoparticles, enhancing the stability (95.4% of the initial capacity was retained after 1000 cycles) [9].

Due to the abovementioned developments, the application of hybrid electrode materials in energy storage has led to remarkable achievements. However, there are still obstacles to be resolved in order to meet the application requirements of high power density and energy density devices, including: (1) the many unanswered fundamental questions about the energy storage mechanisms of electron transport and atomic transport in the interface processes, the answers to which can give us a direction for advancing the performance of the devices; (2) the neglect of the growth mechanism of the hybrid electrode, due to the fact that most research has focused on the performance of the electrodes; and (3) the problem that, although various techniques have been applied to synthesize the electrode materials to date, there is no single method that can meet the requirements of simplicity, green energy, large scale, and cost-effectiveness so as to realize the expected practical application goals.

Nickel oxide (NiO) is regarded as an excellent material for broad use in gas sensors, catalysis, batteries, and supercapacitors due to its theoretical high specific capacity, high chemical/thermal stability, and the fact that it is environmentally friendly and inexpensive [10–13]. Up to now, the differences in the morphology and structure of the NiO materials (nanoflowers, nanoparticles, nanoplates, and mesoporous structure) have been produced by various methods [14–17]. However, these materials have limitations, including a low specific capacity for practical application in supercapacitors, poor conductivity, a propensity for aggregation during the preparation process, and structural instability. Presently, two approaches are widely used to solve these problems: doping foreign atoms, and hybridization with carbonaceous nanomaterials [18–21]. $NiO_x$@graphene composites, as electrode materials, have many advantages, such as their simple preparation process, good stability, low cost, and high specific capacitance [22–24]. $NiO_x$@graphene is a sound model system that can be used to explore the growth and charge storage mechanisms of the hybrid electrode materials. In this work, we attempt to revisit $NiO_x$@graphene composite materials and to explore the growth mechanisms and synergistic effects of the hybrid materials, providing experimental evidence to further improve the electrochemical performance of hybrid electrode materials in the future. A simple, reproducible, affordable, and fast microwave heating method was used to anchor the $NiO_x$ to the surface of the graphene, due to its time-saving advantages, capabilities of uniform heating, and controllable process. The electrochemical behavior of the $NiO_x$@graphene composites was investigated using cyclic voltammetry (CV), galvanostatic charge–discharge (GCD), and electrochemical impedance spectroscopy (EIS) tests. The structure and morphology of the $NiO_x$@graphene composites were characterized by X-ray diffraction (XRD) and scanning electron microscopy (SEM).

## 2. Experimental

### 2.1. Material Characterization

Scanning electron microscopy (Thermo Scientific Apreo FE-SEM, Waltham, MA, USA) and transmission electron microscopy (TEM, FEI Tecnai F20, Lausanne, Switzerland) were used to determine the morphology and microstructure of the powder samples. Powder X-ray diffraction (PXRD) patterns with a Cu target (45 kV, 40 mA) were carried out on Philips *X'*pert MPD (Philips, Eindhoven, Netherlands) in the $2\theta$ range from 20 to 80, with a scan speed of $0.06°$/min to resolve the phase and crystal structures of the materials. The electrochemical performance was measured using a CH Instrument (CHI 760D, Austin, TX, USA) potentiostat with 'Electrochemical Analyzer' software (version 15.03). The stability tests were performed using an Arbin Instrument (version 4.21). The chemical composition of the powder samples was investigated by energy-dispersive X-ray spectroscopy (EDS, EDAX Instruments, Thermo Scientific Apreo FE-SEM, Waltham, MA, USA). The elemental valence state of the thus-prepared samples was determined by X-ray photoelectron spectroscopy (XPS, Kratos XSAM 800, Thermo Scientific Apreo FE-SEM, Waltham, MA, USA).

### 2.2. Materials

All chemicals and reagents were used without further treatment or purification. *N,N*-dimethylformamide (DMF) was supplied by Macron Fine Chemicals (Sanborn, NY, USA). Poly (vinylidene fluoride) (PVDF), carbon black, and nickel (II) hydroxide ($Ni(OH)_2$) were prepared according to the previous literature [25]. A nickel (II) acetate tetrahydrate ($Ni(Ac)_2 \cdot 4H_2O$) and nickel nitrate hexahydrate ($Ni(NO_3)_2 \cdot 6H_2O$) were offered by Alfa Aesar (Burlington, NJ, USA). Sodium hydroxide was purchased from TCI America (KITA-KU, Tokyo, Japan). Graphene was acquired from Magnolia Ridge Inc. (Magnolia, TX, USA)

### 2.3. Preparation of the NiO$_x$/Graphene Nanocomposite

A certain proportion of metal precursors ($Ni(OH)_2$, $Ni(Ac)_2 \cdot 4H_2O$, or $Ni(NO_3)_2 \cdot 6H_2O$) and graphene (25 mg) were added into a glass vial and mixed with a high-speed mixer at a speed of 2000 rpm for 2 min. Subsequently, the mixture was transferred to a domestic microwave oven (Panasonic NN-SN733B, 2.45 GHz, power 1250 W). Finally, the reactor was spontaneously cooled to room temperature and the product of the NiO$_x$@graphene composite was collected. During the microwave irradiation process, graphene acts as a good microwave absorber and thermally conductive layer, providing heat to promote the conversion of the metal precursor to metal oxide. The formation mechanism of NiO$_x$ can be described as follows. Firstly, the metal precursor nanoparticles undergo intense heating when microwave energy is applied, resulting in molecular collisions. At this stage, the amorphous nuclei are formed, but the size of the nanoparticles are small. Following this stage, aggregation/self-assembly is caused by van der Waals interactions, leading to larger particles. Finally, the aggregate directional growth forms a specific crystal orientation and morphology due to differences in the respective surface energies. Scheme 1 illustrates the process of the synthesis of the NiO$_x$@graphene composites. In order to study the influence of the reaction conditions on the electrochemical performance, the type of metal precursor and feeding ratio between the nickel precursor and graphene were accurately controlled. The details are given in Tables S1–S3.

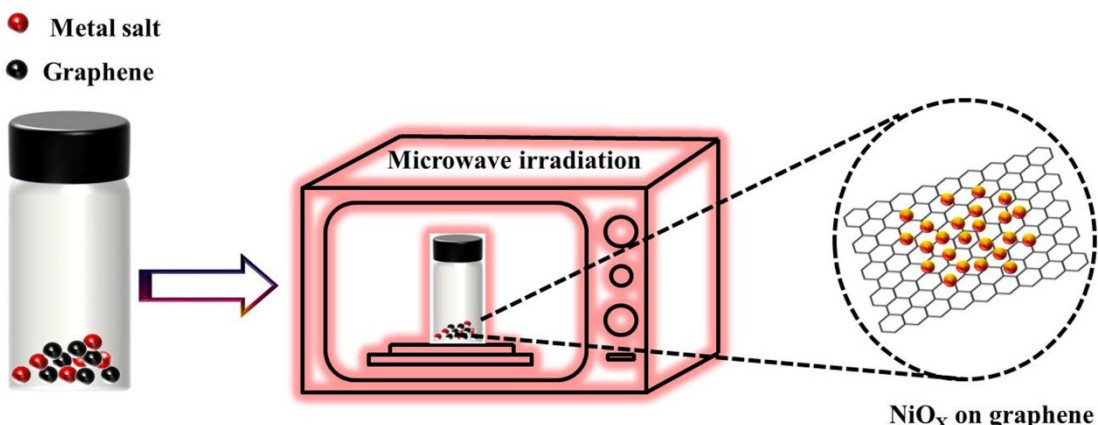

**Scheme 1.** Illustration of the microwave synthesis of the NiO$_x$@graphene composites.

*2.4. Electrochemical Measurements*

The electrochemical performance was investigated using galvanostatic charge–discharge (GCD), cyclic voltammetry (CV), and electrochemical impedance spectroscopy (EIS) tests in three electrode cells. The CV curves were measured at a potential window of 0–0.6 V under different scan rates (5–50 mV/s), and GCD tests were conducted at a potential of 0–0.5 V with different current densities (0.5–5 A/g). The EIS analysis was tested with an open-circuit voltage using an AC amplitude of 0.005 V in the frequency range of $10^{-1}$ to $10^5$ Hz. The working electrode was fabricated by mixing together the active material, carbon black, and polyfluortetraethylene (PVDF) in an 80:10:10 weight ratio. Then, *N,N*-dimethylformamide (DMF) was added to the mixture as a solvent to form the homogeneous slurry, which was coated onto the pretreated nickel foam ($1 \times 1$ cm$^2$). The electrode was dried at 60 °C for 24 h under a pressure of 20 MPa, and the mass loading of the active material was about 2 mg/cm$^2$. The Hg/HgO electrode and platinum sheet ($1 \times 1$ cm$^2$) were used as the reference electrode and counter electrode, respectively. The electrolyte was a 6.0 M KOH aqueous solution. The specific capacitance value was calculated from the GCD curves according to the following equation [26]:

$$C = I\Delta t/m\Delta V \tag{1}$$

where C is the specific capacitance (F/g); m is the mass loading of the active material in the working electrode (mg); I is the constant current (A); $\Delta t$ is the discharge time (s); and $\Delta V$ is the potential window (V). When the mass in the formula was replaced by the area of the electrode (cm$^2$), the areal specific capacitance (C$_A$, F/cm$^2$) was obtained.

## 3. Results and Discussion

The crystal structures of the NiO$_x$@graphene composites were characterized by PXRD. As shown in Figure 1, the peak centered at 2θ = 30.92 in the XRD patterns corresponds to the (002) reflection of graphene (JCPDS card 89-7231). The diffraction peaks appearing at 2θ = 43.5, 50.7, and 74.5 correspond to the (110), (−111), and (111) planes of NiO (JCPDS card 65-7425), respectively. The peaks marked with asterisks correspond to Ni$_2$O$_3$ (JCPDS card 14-0481), indicating that different products were produced when different metal precursors were used. Furthermore, the diffraction peaks of NiO$_x$@GR-A2 were markedly broader than those of NiO$_x$@GR-B2 and NiO$_x$@GR-C2, indicating a small crystallite size [27]. The average crystallite dimensions estimated by the Scherrer equation [28], based on the (200) reflections, are approximately 1.398, 5.935, and 4.202 Å, respectively. The oxidation states of the NiO$_x$ nanoparticles were measured by XPS. In the full survey scan spectrum (Figure S1a), the C 1s and O 1s peaks located at 285.6 eV and 530.4 eV corresponded to the graphene and the oxygen in NiO, respectively [29]. In the high resolution XPS spectra of Ni 2p (Figure S1b), the two major peaks appeared at 855.5 eV (Ni 2p$_{3/2}$) and 873.2 eV

(Ni $2p_{1/2}$), with two satellite peaks at 861.5 eV and 880.3 eV, respectively, indicating the existence of $Ni_2O_3$ and NiO in the composite [30,31].

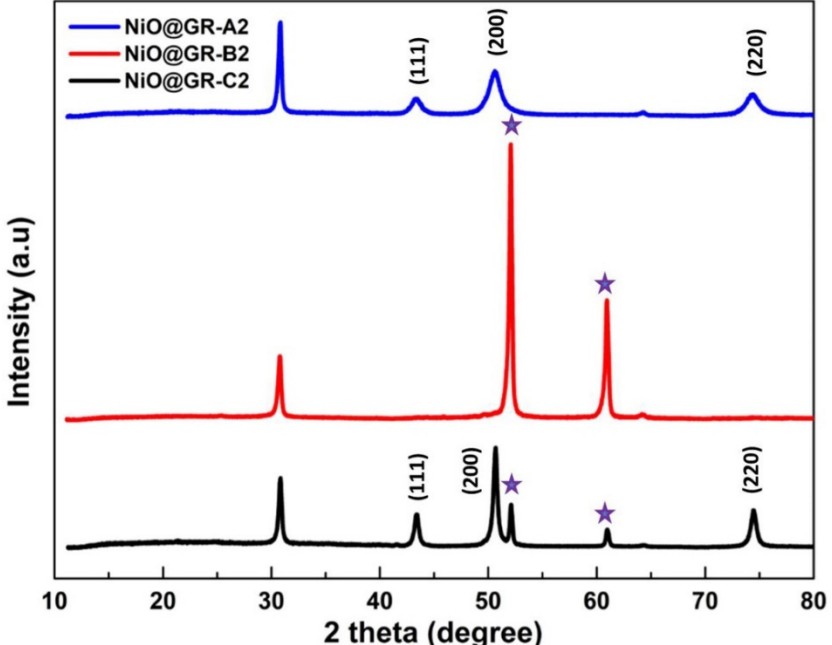

**Figure 1.** XRD patterns for $NiO_x$@GR-A2, $NiO_x$@GR-B2, and $NiO_x$@GR-C2.

The microstructure and morphology of $NiO_x$@graphene were characterized by SEM and TEM. Figure 2 shows the SEM and TEM images of the $NiO_x$@graphene composites prepared using the different metal precursors, ($Ni(OH)_2$, $Ni(Ac)_2 \cdot 4H_2O$, or $Ni(NO_3)_2 \cdot 6H_2O$). When $Ni(OH)_2$ acted as the precursor, the $NiO_x$ exhibited a bulk form with a crumpled surface, as shown in Figure 2a,b. When the $NiO_x$@graphene was prepared using $Ni(Ac)_2 \cdot 4H_2O$ and $Ni(NO_3)_2 \cdot 6H_2O$ as the precursors, the corresponding $NiO_x$ exhibited a good flower-like structure (Figure 2d,e) and small particles with a fluffy surface (Figure 2g,h), individually. The TEM image further illustrates the porous and hollow structures of the $NiO_x$ composites (Figure 2c,f,i). The different interior structures of the $NiO_x$ composites may arise from the different decomposition temperatures and crystal structures of the metal precursors. The decomposition processes of $Ni(Ac)_2 \cdot 4H_2O$ and $Ni(NO_3)_2 \cdot 6H_2O$ generate gas, which is beneficial for the formation of the porous structures. Among the three metal precursors, $Ni(NO_3)_2 \cdot 6H_2O$ has the highest content of crystallization water, which hinders the absorption of microwave energy by the graphene and slows down the heating process. This leads to the creation of small-sized particles that provide a larger effective contact area. It is clear that the morphology and nanostructure of $NiO_x$ is metal precursor-dependent. Furthermore, the EDS elemental mapping of $NiO_x$@GR-C2 showed a uniform distribution of C, Ni, and O (Figure S2), which indicates that $NiO_x$ was homogeneously distributed throughout the composite. As is well-known, the uniform distribution of the reacting sites is a key factor in ensuring electrode stability and electron transfer, which helps to avoid unnecessary resistance and energy loss.

By controlling the reaction conditions (the type of metal precursor and feeding ratio between the nickel precursor and graphene), we can better examine the trends of the electrochemical properties of the materials. The specific capacitance was calculated based on the GCD curves. Figure 3 indicates the dependence of the specific capacitance of the $NiO_x$@graphene nanocomposites prepared using different metal precursors at different current densities. With the increase in the feeding ratio of the nickel precursor to graphene, the specific capacitance initially increased and then decreased. The reason for this result is probably the increased content of generated $NiO_x$ after the decomposition of the nickel precursor. However, as the amount of added metal precursor increased, the energy required

for the complete decomposition of the metal precursor also increased accordingly. Based on the graphene content supplied in this case (25 mg), the amount was not sufficient to produce enough energy to completely decompose all the metal precursors, and finally resulted in the decrease in the electrochemical properties.

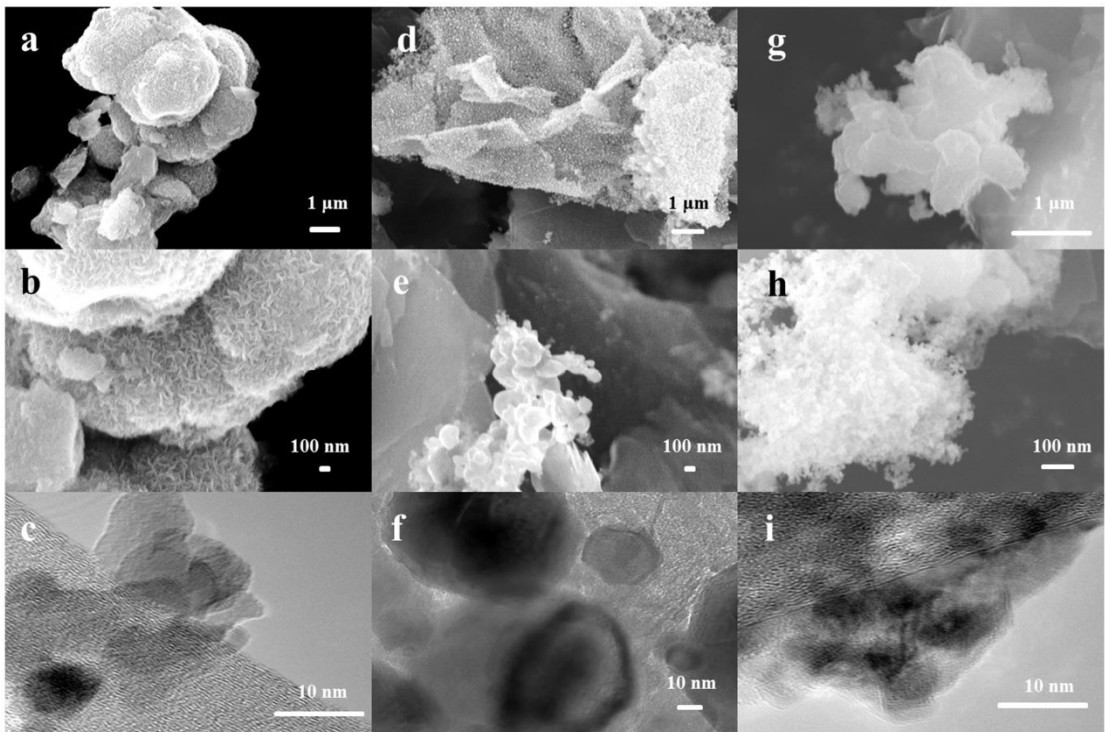

**Figure 2.** SEM images of (**a**,**b**) NiO$_x$@GR-A2, (**d**,**e**) NiO$_x$@GR-B2, and (**g**,**h**) NiO$_x$@GR-C2 at different magnifications. TEM images of (**c**) NiO$_x$@GR-A2, (**f**) NiO$_x$@GR-B2, and (**i**) NiO$_x$@GR-C2.

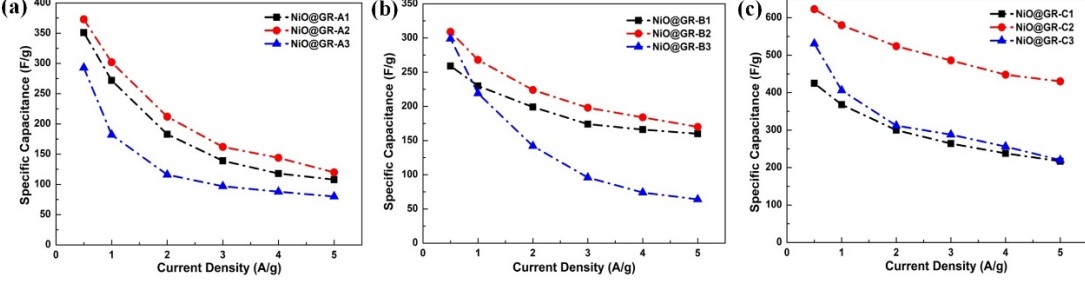

**Figure 3.** Specific capacitance of the NiO$_x$@GR nanocomposites at different current densities: (**a**) NiO$_x$@GR-A1, NiO$_x$@GR-A2, and NiO$_x$@GR-A3; (**b**) NiO$_x$@GR-B1, NiO$_x$@GR-B2, and NiO$_x$@GR-B3; and (**c**) NiO$_x$@GR-C1, NiO$_x$@GR-C2, and NiO$_x$@GR-C3.

When comparing the electrochemical performance of the NiO$_x$@GR nanocomposites prepared using the different metal precursors under the optimal ratio (NiO$_x$@GR-A2, NiO$_x$@GR-B2, and NiO$_x$@GR-C2), NiO$_x$@GR-C2 exhibited the highest specific capacitance, corresponding to the largest integral area in the CV curves and the longest discharge time in the GCD curves at the same scan rate and current density (Figure 4a–c). This composite might benefit from the small particles and fluffy surface of the metal oxide increasing the surface area, which can boost the exposed active sites, facilitating contact with electrolytes, and the porous structure, which provides more pathways for ion transport, consequently promoting the faraday reaction. The effects of the metal precursors on the cyclic stability of the composite were also investigated at the current density of 1 A/g (Figure 4d). It can be seen that the capacitance retentions were 72.8%, 58.6%, and 62.2% for NiO$_x$@GR-A2, NiO$_x$@GR-B2, and NiO$_x$@GR-C2, respectively, after 4000 cycles in each case. The relatively

high capacitance retention might be due to the synergistic effect between the graphene and the metal oxide, which benefited from the buffering of the mechanical stress during the charge–discharge processes and increased the redox efficiency [32,33]. Among the prepared composites, NiO$_x$@GR-A2 presented with the highest cycle stability, which might be related to the bulk form and crumpled surface of the NiO$_x$. Moreover, in Table 1, the electrochemical properties of NiO$_x$@GR prepared through microwave heating are compared with previously reported approaches. It is obvious that the microwave heating method shortens the reaction times from hours to minutes and exhibits a higher specific capacitance. The improved capacitance of NiO$_x$@GR may be attributed to the strong interaction between the NiO$_x$ and the graphene sheets, resulting in an enhanced charge transport [34].

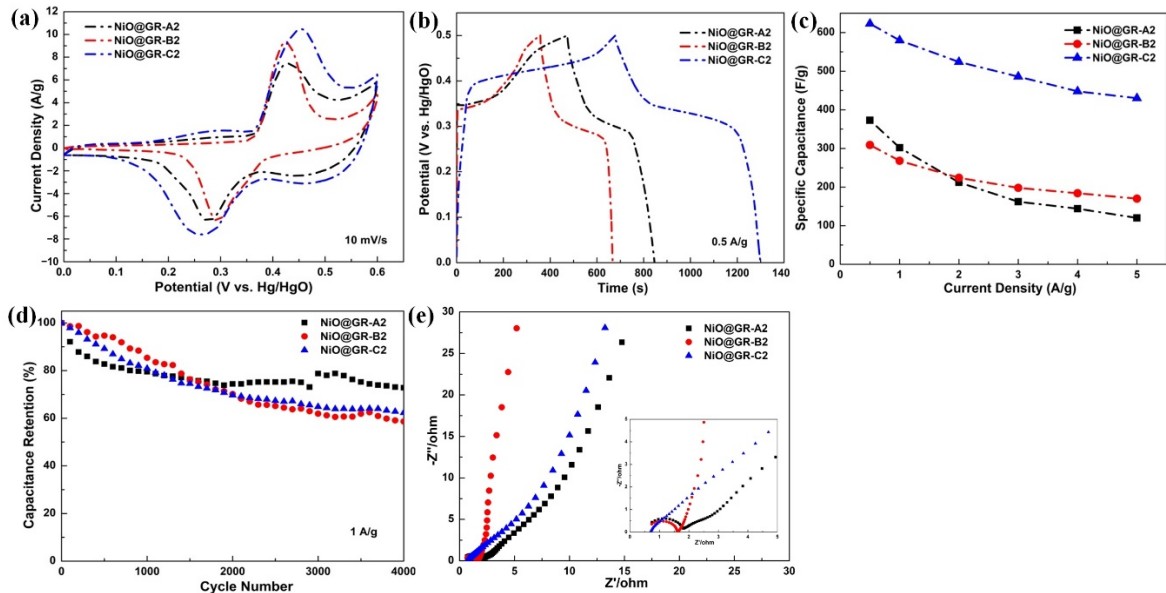

**Figure 4.** (**a**) CV and (**b**) GCD curves of NiO$_x$@GR-A2, NiO$_x$@GR-B2, and NiO$_x$@GR-C2. (**c**) Specific capacitance of the NiO$_x$@GR-A2, NiO$_x$@GR-B2, and NiO$_x$@GR-C2 nanocomposites at different current densities. (**d**) Cyclic stability of NiO$_x$@GR-A2, NiO$_x$@GR-B2, and NiO$_x$@GR-C2. (**e**) Nyquist plots of the EIS for the NiO$_x$@GR-A2, NiO$_x$@GR-B2, and NiO$_x$@GR-C2 electrodes. The inset shows the enlarged EIS of the electrodes.

EIS analysis was conducted to investigate the electrochemical characteristics between the electrode and the electrolyte interface. The Nyquist plots consist of two parts: a small semicircle in the high-frequency region, and a straight line in the low-frequency region. In general, the semicircle is presented according to the charge transfer resistance of the electrode and the diameter of the semicircle is equal to the electrode resistance (R$_{ct}$). The vertical line corresponding to the Warburg impedance reflects the capacitive behavior of the electrode [35–37]. NiO$_x$@GR-C2 exhibited a smaller semicircle diameter than NiO$_x$@GR-A2 and NiO$_x$@GR-B2 in the high-frequency region, which indicates that it had a lower charge transfer resistance. Furthermore, the NiO$_x$@GR-B2 electrode presented a nearly vertical linear shape in the low-frequency region, indicating that it had a better capacitive behavior than the other two electrodes due to the rapid ion diffusion (Figure 4e).

Figure 5 shows the CV curves of the NiO$_x$@GR-C2 electrode in the potential range of 0 to 0.6 V at various scan rates. A pair of oxidation/reduction peaks can be clearly observed, corresponding to the pseudocapacitance of the electrode. This was mainly due to the faradaic conversions between Ni (II) and Ni (III) in an alkaline medium, which can be elaborated by the following reaction:

$$NiO + OH^- \leftrightarrow 2NiOOH + e^- \tag{2}$$

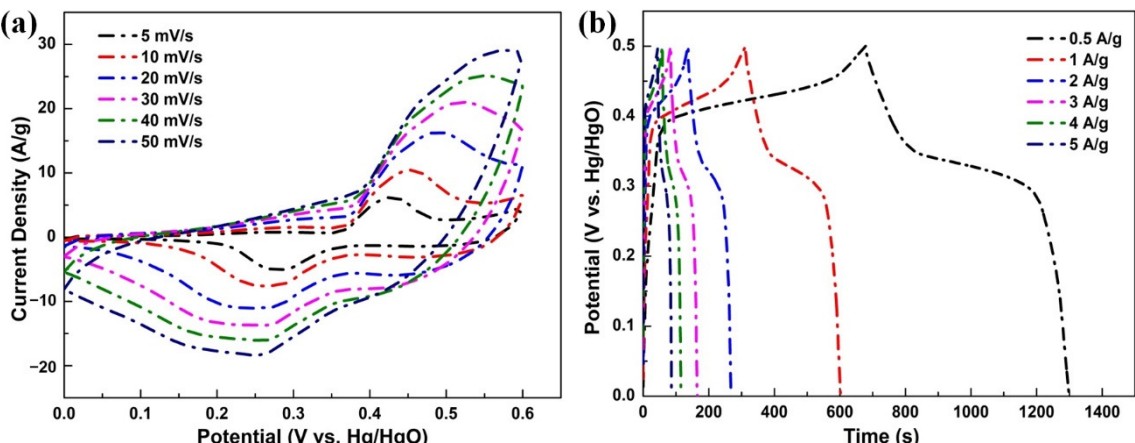

**Figure 5.** (**a**) CV curves of $NiO_x$@GR-C2 at varied scan rates. (**b**) GCD curves of $NiO_x$@GR-C2 at different current densities.

As the scan rate increased, a slight shift in the peaks was observed, which may be ascribed to the polarization effect of the electrode [38]. In addition, the shape of the CV curves continued to indicate that the electrode had a good rate capability (Figure 5a) [39]. The GCD curves presented with a similar plateau, matching the redox peaks of the CV curves (Figure 5b). As the current density decreased, the discharge time increased corresponding to the increase in the specific capacitance of the electrode. It can be concluded that, at a low current density, the charge had enough time to diffuse into the inner core of the electrode and access the active sites. Comparatively, at a higher current density, the current density increased rapidly, which caused a large number of ions to gather at the solid/liquid interface, increasing the internal diffusion resistance of the electrode [40,41]. From 0.5 to 5 A/g, the specific capacitance of $NiO_x$@graphene shifted from 623 F/g (1.24 $F/cm^2$) to 430 F/g (0.86 $F/cm^2$). This value was higher than those previously reported for pure NiO [9,42,43]. The reason for the good rate capability and high capacitance of $NiO_x$@graphene might be the combination of the characteristics of metal oxide and graphene. The highly conductive graphene with a high area surface not only offered high electrical conductivity, but also provided more active sites and more interfacial contact for the redox reaction. The $NiO_x$ with a highly porous structure offered more charge transfer channels. The increased contact area and evenly distributed porous structure can shorten the ion migration pathway and facilitate the transportation of electrons. Moreover, graphene, as a carrier, can give the metal oxides physical support, increasing the stability of the nanocomposites during the charging and discharging processes [44,45].

To further gain insight into the reaction mechanisms, we can distinguish the effect of the capacitance on the electrode by kinetic analysis of the CV curves at various scan rates. The capacitive effect of the electrode can be described by plotting the relationship between the peak current and the sweep rate, with the following equation [46]:

$$i = av^b \tag{3}$$

where a and b are variable constants. By determining the value of b, we can determine the contribution of the diffusion control and capacitive effect to the total capacitance. When the b value is close to 0.5, the total capacitance is dominated by the diffusion-controlled faradaic process. When the b value is close to 1, the charge storage process is controlled by the capacitive effect. As the value of b is between 0.5 and 1, this proves that the electrochemical process benefits from both the capacitive and diffusion-limited redox processes. The contributions of the capacitive effect and diffusion-controlled contribution to the total capacitance can be quantified using Dunn's method [47]:

$$i_{total} = k_c v + k_d v^{0.5} \tag{4}$$

where $k_c v$ is the capacitive effect and $k_d v^{0.5}$ is the diffusion-controlled contribution. Then, transforming the equation into:

$$i_{total}/v^{0.5} = k_c v^{0.5} + k_d \qquad (5)$$

through plotting $i_{total}/v^{0.5}$ versus $v^{0.5}$, the $k_c$ and $k_d$ can be determined. As shown in Figure 6b, the b values of the oxidation and reduction peaks were 0.6626 and 0.5544, respectively, suggesting that the charge storage of NiO$_x$@GR-C2 benefited from both the diffusion-controlled contribution and capacitive effect. The capacitive current separated from the total measured currents under the scan rate of 20 mV/s is displayed in Figure 6a. The details of the capacitive effect and diffusion-controlled contribution at different scan rates are shown in Figure 6c. At 5 mV/s, the percentage of the diffusive mechanism contribution was 93.4%. As the scan rate decreased, the capacitive contribution increased, while the diffusive contribution increasingly depressed. This was due to the fact that the ions did not have enough time to intercalate inside the material structures. However, in this system, the percentage of the pseudocapacitance contribution still remained at 90.2% at 50 mV/s; thus, the pseudocapacitance contribution was believed to be dominative.

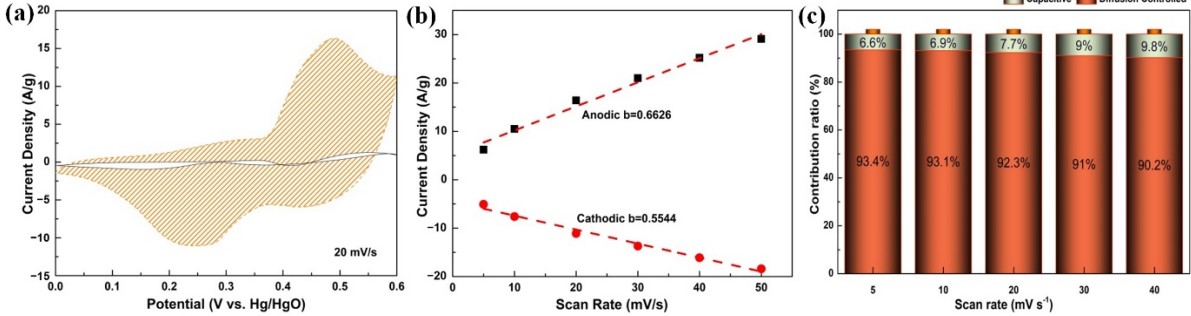

**Figure 6.** (**a**) CV graph indicating the capacitive contribution of NiOx@GR-C2 at 20 mV/s. (**b**) The calculated b values at different scan rates. (**c**) A stacked bar graph showing the percentage of the total capacitance attributed to the diffusion-limited and capacitive contributions.

**Table 1.** Comparison of the electrochemical properties of NiOx@GR prepared through microwave heating with previously reported approaches.

| Method | Specific Capacitance | Stability | Reaction Time | Ref |
|---|---|---|---|---|
| Thermal decomposition | 417 F/g, 13 A/g | 85.5%, 3000 cycles | 10 min | [48] |
| Hydrothermal and thermal decomposition | 430 F/g, 0.2 A/g | 86.1%, 2000 cycles | 4 h | [44] |
| Solvothermal | 587 F/g, 1 A/g | 98%, 1000 cycles | 12 h | [49] |
| Sol-gel | 628 F/g, 1 A/g | 82.4%, 3000 cycles | 24 h | [34] |
| Hydrothermal | 500 F/g, 5 mV/s | 84%, 3000 cycles | 2 h | [43] |
| Microwave heating | 623 F/g, 0.5 A/g | 62.2%, 4000 cycles | 5 min | **Our work** |

## 4. Conclusions

In our work, we reported a simple, reproducible, low-cost, and fast approach to microwave synthesis for the preparation of hybrid electrode architectures. The synthesized nanocomposites were characterized using XRD, TEM, SEM, and EDS methods. The effects of the reaction conditions (the type of metal precursor and feeding ratio between the nickel precursor and graphene) on the formation mechanism of the electrode was demonstrated. The results proved that the microstructure and morphology of the electrode materials were metal precursor-dependent, which was related to the electrochemical performance of the electrodes. In addition, kinetic analysis was used to gain insight into the charge storage mechanisms at the interface between the electrode and the electrolyte during the energy storage process, which can give us a direction for advancing the performance of the devices. This work can serve as a model for understanding the growth mechanisms and the synergistic effects of hybrid electrode materials consisting of carbon-based materials

and metal oxides, and offers experimental support for the designing of hybrid electrode materials with an excellent electrochemical performance in the future.

**Supplementary Materials:** The following supporting information can be downloaded at: https://www.mdpi.com/article/10.3390/coatings12081060/s1, Figure S1: XPS spectrum of the NiOx@GR-C2; (b) High-resolution XPS spectrum of Ni 2p; Figure S2: (a) SEM image of NiOx@GR-C2 (inset: EDS composite elemental map); elemental mapping: (b) C, (c) Ni, (d) O; (e) EDS spectrum; Table S1: Operation conditions for synthesizing NiOx@graphene; Table S2: Operation conditions for synthesizing NiOx@graphene; Table S3: Operation conditions for synthesizing NiOx@graphene.

**Author Contributions:** Conceptualization, X.Z.; Data curation, Y.L.; Formal analysis, Y.L.; Investigation, Y.L. and Z.W.; Methodology, Y.L.; Project administration, X.Z.; Resources, R.W. and X.Z.; Supervision, X.Z.; Validation, Y.L.; Visualization, Y.L.; Writing—original draft, Y.L.; Writing—review & editing, Y.L., Z.W., R.W. and X.Z. All authors have read and agreed to the published version of the manuscript.

**Funding:** This research received no external funding.

**Acknowledgments:** Y. Liang acknowledges the financial support of the China Scholarship Council (No. 201708510080).

**Conflicts of Interest:** The authors declare no conflict of interest.

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
