# Peer review of "The Microwave Facile Synthesis of NiOx@graphene Nanocomposites for Application in Supercapacitors: Insights into the Formation and Storage Mechanisms"

_coatings, doi:10.3390/coatings12081060_

Round 1
Reviewer 1 Report
The paper describes promising composite material and detailed investigation of properties and functionality.
I have a few suggestions to improve the manuscript.
What is AC in the abbreviation “NiO//CNT//AC“?
Please, show at the XPS graphs of Ni2p the names of the bonds (oxidation states) that correspond to the fitted peaks.
Line 149: Please, mention the XPS spectrum calibration to C-C peak (285eV) according to the literature [DOI: 10.3390/MA14061428].
Was it possible to define NiOx grain-size from XRD?
Please, number the chemical equations.
Please, correct graph at Fig. 4d
Author Response
Dear Editors and Reviewers:
Thanks very much for the comments and suggestions concerning our manuscript. Those comments are all valuable and very helpful to revise and improve our paper, as well as the important guiding significance to our future researches. We have studied comments carefully and have made correction which we hope meet with approval. Revised portion are marked in Red/Blue in the revised manuscript. The main corrections in the paper and the responds to the comments are shown as flowing:
Reviewer 1 (Response with RED color in the revised manuscript)
The paper describes promising composite material and detailed investigation of properties and functionality.
Response: We highly appreciate your positive feedback. We have carefully revised our manuscript following your comments and suggestions. Our detailed responses are given below point-by-point.
Comments (1): What is AC in the abbreviation “NiO//CNT//AC”?
Responses (1): Thanks for your kind comments. The AC in the abbreviation “NiO//CNT//AC” is activated carbon. We feel very sorry for our negligence and we have replaced “NiO//CNT//AC” with “NiO-CNT//Activated Carbon” on page 2.
Comments (2): Please, show at the XPS graphs of Ni 2p the names of the bonds (oxidation states) that correspond to the fitted peaks.
Responses (2): Thanks for your kind comments. We have marked the oxidation states of Ni 2p in Fig S1.
Comments (3): Line 149: Please, mention the XPS spectrum calibration to C-C peak (285eV) according to the literature [DOI: 10.3390/MA14061428].
Responses (3): Thanks for your kind comments. We have mentioned the C-C peak (285 eV) in the XPS spectrum according to the literature [DOI: 10.3390/MA14061428] (Page 4, line 161-163).
Comments (4): Was it possible to define NiOx grain-size from XRD?
Responses (4): Thanks for your kind comments. According to your suggestion, we have calculated the average crystallite dimension by the Scherrer equation based on the (200) reflections.
Where K-shape factor between 0.9 and 1.1, λ-incident X-ray wavelength (here λCuKα=1.542 Å), β-full width at half-maximum (FWHM) of a given line and θ-position of that line in the pattern.
The crystallite sizes are about 1.398, 5.935, and 4.202 Å for NiOx@GR-A2, NiOx@GR-B2 and NiOx@GR-C2, respectively. We have added these results in the revised manuscript (Page 4, line 157-160).
Comments (5): Please, number the chemical equations.
Responses (5): Thanks for your kind comments. We have numbered the chemical equations with red color in the revised manuscript.
Comments (6): Please, correct graph at Fig. 4d
Responses (6): Thanks for your kind comments. We have carefully checked Fig. 4d, but not sure what is needed to be corrected. We just improved the resolution of Fig.4. If there is anything else that we could do, please do not hesitate to let us know.

Reviewer 2 Report
Manuscript ID: coatings-1827514
General comments:
This manuscript Microwave Facile Synthesis of NiOx@graphene Nanocomposites for Application in Supercapacitors: Insights into the Formation and Storage Mechanism. The present study illustrates the production of NiOx@graphene nanocomposites using microwave heating method. The produced composites were used as a model system to explore the growth and charge storage mechanisms of the hybrid electrode materials due to the simple preparation process, good stability, low cost, and high specific capacitance.
The work is adequate. The comments may be useful for the improvement of the manuscript. Minor revisions are needed to make the work acceptable.
Some specific comments are as follows:
1. Some recent relevant works should be cited in the introduction.
2. In abstract one to two lines should be added to explain the statement of problem.
3. The specific area measurement must be included to support the results.
4. Improve the quality of Figures 3-5.
5. The English of the manuscript should be revised.

Author Response
Reviewer 2 (Response with Blue color in the revised manuscript)
This manuscript Microwave Facile Synthesis of NiOx@graphene Nanocomposites for Application in Supercapacitors: Insights into the Formation and Storage Mechanism. The present study illustrates the production of NiOx@graphene nanocomposites using microwave heating method. The produced composites were used as a model system to explore the growth and charge storage mechanisms of the hybrid electrode materials due to the simple preparation process, good stability, low cost, and high specific capacitance.
The work is adequate. The comments may be useful for the improvement of the manuscript. Minor revisions are needed to make the work acceptable.
Response: We highly appreciate your positive feedback. We have carefully revised our manuscript following your comments and suggestions. Our detailed responses are given below point-by-point.
Comments (1): Some recent relevant works should be cited in the introduction.
Responses (1): Thanks for your kind comments. We have cited some recent relevant works in the introduction (Page 2, line 73-74).
- Liu, Y.; Gao, C.; Li, Q.; Pang, H., Nickel oxide/graphene composites: synthesis and applications. Chem. Eur. J. 2019, 25(9), 2141-2160.
- Bu, Y.; Wang, S.; Jin, H.; Zhang, W.; Lin, J.; Wang, J., Synthesis of porous NiO/reduced graphene oxide composites for super-capacitors. J. Electrochem. Soc. 2012,159 (7), A990.
- Hui, X.; Qian, L.; Harris, G.; Wang, T.; Che, J., Fast fabrication of NiO@graphene composites for supercapacitor electrodes: Combination of reduction and deposition. Materials & design. 2016,109, 242-250.
Comments (2): In abstract one to two lines should be added to explain the statement of problem.
Responses (2): Thanks for your kind comments. Based on your suggestion, we have added the description of the problem in the abstract (Page 1, line 10-13).
Comments (3): The specific area measurement must be included to support the results.
Responses (3): Thanks for your kind comments. We have added the areal specific capacitance with blue color to support the results (line 148-149, line 265-266).
Comments (4): Improve the quality of Figures 3-5.
Responses (4): Thanks for your kind comments. We have improved the quality of Figures 3-5 in the revised manuscript.
Comments (5): The English of the manuscript should be revised.
Responses (5): Thanks for your kind comments. We have checked and corrected the grammar and spelling with blue color in the revised manuscript.

Reviewer 3 Report
Dear authors,
1. In my opinion, in the introduction it is necessary to add not only the areas of application of nickel as a catalytically active material and compositions based on it, but also talk about alternative methods for its synthesis and deposition on the reaction surface. For example
- 10.1039/C3TA12736B
- 10.1039/C3RA23286G
- 10.1016/j.ceramint.2020.06.318
- 10.1149/1.1836396
Other than that everything is great
Author Response
We highly appreciate your positive feedback. We have carefully revised our manuscript following your comments and suggestions and added the corresponding description in the introduction (page 2)
